# Don't judge a book or health app by its cover: User ratings and downloads are not linked to quality

Maciej Hyzy[1,2]*, Raymond Bond[1], Maurice Mulvenna[1], Lu Bai[3], Anna-Lena Frey[2], Jorge Martinez Carracedo[1], Robert Daly[2], Simon Leigh[2,4]

1 School of Computing, Ulster University, Belfast, United Kingdom, 2 ORCHA, Sci-Tech Daresbury, Violet V2, Daresbury, United Kingdom, 3 School of Electronics, Electrical Engineering and Computer Science, Queen's University Belfast, Belfast, United Kingdom, 4 Warwick Medical School, University of Warwick, Coventry, United Kingdom

* maciejmarekzych@gmail.com

## Abstract

### Objective

To analyse the relationship between health app quality with user ratings and the number of downloads of corresponding health apps.

### Materials and methods

Utilising a dataset of 881 Android-based health apps, assessed via the 300-point objective Organisation for the Review of Care and Health Applications (ORCHA) assessment tool, we explored whether subjective user-level indicators of quality (user ratings and downloads) correlate with objective quality scores in the domains of user experience, data privacy and professional/clinical assurance. For this purpose, we applied spearman correlation and multiple linear regression models.

### Results

For user experience, professional/clinical assurance and data privacy scores, all models had very low adjusted R squared values (< .02). Suggesting that there is no meaningful link between subjective user ratings or the number of health app downloads and objective quality measures. Spearman correlations suggested that prior downloads only had a very weak positive correlation with user experience scores (Spearman = .084, p = .012) and data privacy scores (Spearman = .088, p = .009). There was a very weak negative correlation between downloads and professional/clinical assurance score (Spearman = -.081, p = .016). Additionally, user ratings demonstrated a very weak correlation with no statistically significant correlations observed between user ratings and the scores (all p > 0.05). For ORCHA scores multiple linear regression had adjusted R-squared = -.002.

**Data Availability Statement:** All relevant data are within the manuscript and its Supporting information files.

**Funding:** This research has been funded by the department for the economy (DfE) in the UK and by company called ORCHA.

**Competing interests:** I have read the journal's policy and the authors of this manuscript have the following competing interests: This study is funded by a DfE Cast award and ORCHA. Simon Leigh, Anna Frey and Robert Daly are/were employees at ORCHA when this study was conducted. Other authors have no conflict of interest to declare.

## Conclusion

This study highlights that widely available proxies which users may perceive to signify the quality of health apps, namely user ratings and downloads, are inaccurate predictors for estimating quality. This indicates the need for wider use of quality assurance methodologies which can accurately determine the quality, safety, and compliance of health apps. Findings suggest more should be done to enable users to recognise high-quality health apps, including digital health literacy training and the provision of nationally endorsed "libraries".

## Introduction

According to a report from 2021 there were more than 350,000 health apps available in the iOS and Android stores, with an estimated 250 health apps added every day [1]. Moreover, searches for digital health products within app stores have also increased [2]. A potential catalyst for this could have been the COVID-19 pandemic and restricted access to incumbent services. Nevertheless, these findings clearly indicate that the public has an interest in health apps.

However, given the large number of health apps on offer, it can be difficult for users to identify high-quality apps that meet their needs. Notably, selecting the low-quality app can be associated with substantial opportunity costs and/or risks. For example, a systematic assessment of suicide prevention and deliberate self-harm mobile health apps found that some apps encouraged risky behaviours such as the uptake of drugs [3]. Moreover, reviews across different disease areas have shown that many health apps do not comply with data privacy, sharing, and security standards [4–7], have safety concerns [8], provide incomplete or misleading medical information [9,10], lack evidence-based components [11], and/or have not been supported by efficacy/effectiveness studies [5,6,12]. Also, health experts have largely avoided formally recommending apps, which forces users to obtain recommendations from other sources [13]. Therefore, if not sufficiently informed, user's app choices can result in poor health benefits if ineffective apps are chosen and/or pose significant risks to user's health and privacy.

Notably, in the absence of guidance, users are likely to select health apps based on metrics that they perceive to be proxies for quality, such as prior purchases/downloads and user ratings. For instance, a study from 2020 [14], found that besides price, in-app purchase options, and presence of in-app advertisements, user ratings were impactful predictors of user downloads, and the number of downloads increased with average user ratings. However, while metrics such as user ratings may be useful when selecting many other goods and services, they may not accurately reflect the value and risks associated with the use of health apps [15], as these aspects are complex to assess and often not immediately apparent to (prior) users of the app.

In line with this, previous studies have shown that app quality ratings are often not significantly positively associated with user ratings. For instance, user ratings were found not to be significantly correlated with Mobile Application Ratings Scale (MARS) scores [16,17] or 'PsyberGuide credibility ratings scale' (PGCRS) scores [18]. A study from 2022 [19], found a weak but significant negative correlation between their criteria and scores and user ratings for women with anxiety during pregnancy.

These findings suggest that user ratings and downloads are not a good proxy for overall app quality. However, most frameworks are not all-encompassing [20–23], for example, the MARS doesn't include privacy questions. Hence, from the previous findings, it is unclear whether user ratings and download rates may be associated with compliance with quality components,

such as user experience (UX), professional/clinical assurance (PCA) and data privacy (DP). The current study aimed to examine this relationship.

Specifically, this study's objective is to analyse the relationship between health app quality scores (UX, PCA and DP) with user ratings and the number of downloads of corresponding health apps. This study has one hypothesis, user ratings and number of downloads are inadequate predictors of user experience, professional/clinical assurance, and data privacy of health apps.

## Materials and methods

### The dataset provenance

The dataset used for this study was provided by the Organisation for the Review of Care and Health Applications (ORCHA). ORCHA is a United Kingdom (UK) based digital health compliance company that specialises in the assessment of health apps. ORCHA provides an 'ORCHA library' that contains information about health apps that have been assessed regarding professional/clinical assurance, data privacy and user experience, allowing consumers and clinical professionals to make informed decisions whether to use or recommend these health apps. ORCHA is currently working with 70% of the National Health Service (NHS) organisations within England.

ORCHA has provided a dataset comprising 2127 health app assessments which were assessed using the ORCHA Baseline Review tool, Version 6 (OBR V6) [24]. For this study 881 Android health apps have been used, the steps involved in the inclusion of health apps can be found in Fig 1 of S1 Appendix. The OBR V6 tool is the latest version of the 'ORCHA assessment tool' which consists of ~300 objective assessment questions (where most questions are objective dichotomous–yes/no questions). OBR V6 assesses three aspects of a health app, namely 1) professional/clinical assurance (PCA), 2) data privacy (DP), and 3) user experience (UX) (also referred to as 'usability and accessibility'). Each of these three domains is scored individually on a scale from 0 to 100 and these three domain scores are combined into an overall ORCHA score.

The dataset consists of the aggregated user ratings, number of downloads and quality scores (UX, PCA and DP scores) for each health app. Each assessment of the 881 health apps has been carried out by at least 2 trained reviewers, where in the case of a dispute, a third reviewer would resolve it. All reviewers have undergone the same training to use the OBR V6 assessment tool. The dataset used included health app assessments that were published between 18[th] January 2021 and 6[th] January 2022.

### Statistical analysis and modelling

Data was accessed and analysed between July and December 2022. We carried out secondary data analyses of this ORCHA dataset, using R studio and R programming language. Spearman correlations were used to examine how correlated ORHCA, UX, PCA and DP scores are with user ratings (a 1–5 ratings) and number of downloads. The number of downloads variable was converted into download levels, as only download ranges, not exact numbers of downloads, were available. There were 20 ranges of downloads, and each was assigned a download level going from 1 (the smallest) to 20 (the highest). For the analysis the smallest value in each of the 20 ranges was also used as an alternative to the download levels. This was done to improve rigour of the analysis by using two approaches to estimate number of downloads from the available range of downloads.

Multiple linear regression (MLR) was used to model the relationship between app quality scores and the apps' user ratings and downloads. R squared and adjusted R squared metrics were used to measure the fitness of the models. For all statistical tests, a p-*value* < .013

(Bonferroni-corrected for multiple hypothesis testing) was considered statistically significant. If there are any links among user ratings and downloads, and quality scores they should be revealed by spearman correlations and/or MLR.

### Ethical approval

This secondary data analytics study was approved by Ulster University (ethics filter committee for Faculty of Computing, Engineering and the Built Environment). The process undertaken by ORCHA ensures that health app developers are aware of their score and are given time to contest findings of the assessment which may be amended if developers provide additional relevant information. All reviews, unless explicitly asked to be removed by the developer, are covered as suitable for research in ORCHA's privacy policy.

## Results

There was a total of 881 Android health apps used for this study. The categories of health apps and sample size (n) used in this study are depicted in Table 1 in descending order of sample size. Each health app has been assigned to one or multiple categories.

Table 2 depicts sample size, median and interquartile range (IQR) for each score (ORCHA, UX, PCA and DP), most common download level when separated by user ratings (in the intervals of $> = 1$ and $<2$, $> = 2$ and $<3$, $> = 3$ and $<4$, $> = 4$ and $< = 5$). Table 3 depicts ORCHA

**Table 1. Categories and sample size.**

|  | Category | Sample size (n) |
|---|---|---|
| 1 | Healthy living | 319 |
| 2 | Mental health | 253 |
| 3 | Medicines and clinical reference | 88 |
| 4 | Neurological | 68 |
| 5 | Diabetes | 49 |
| 6 | Pregnancy | 47 |
| 7 | Respiratory | 44 |
| 8 | Women's health | 38 |
| 9 | Children's health | 37 |
| 10 | Musculoskeletal disorders | 31 |
| 11 | Cancer | 28 |
| 12 | Sexual health | 33 |
| 13 | Utilities and Administration | 30 |
| 14 | Neurodiverse | 27 |
| 15 | Ophthalmology | 24 |
| 16 | Pain management | 22 |
| 17 | Cardiology | 21 |
| 18 | Dental | 19 |
| 19 | Dermatology | 17 |
| 20 | Gastrointestinal | 15 |
| 21 | Ear/Nose/Throat/Mouth | 14 |
| 22 | First aid | 9 |
| 23 | Social support network | 8 |
| 24 | Urology | 7 |
| 25 | Allergy | 7 |
| 26 | Older adult | 5 |

**Table 2. User ratings details.**

| User ratings | Sample size (Number of apps) | ORCHA score–median (IQR) | UX score–median (IQR) | PCA score–median (IQR) | DP score–median (IQR) | Most common download level |
|---|---|---|---|---|---|---|
| > = 1 and <2 | 8 | 59(7.5) | 72.19(8.96) | 48.16(23.30) | 65(14.36) | 8,10 |
| > = 2 and <3 | 43 | 59(22) | 76.67(11.27) | 46.66(43.50) | 67.44(21.04) | 8 |
| > = 3 and <4 | 222 | 63.5(21) | 75.21(9.42) | 55.67(44.28) | 65.23(17.51) | 10 |
| > = 4 and < = 5 | 608 | 60(24) | 74.55(9.58) | 46.93(46.30) | 65.36(18.77) | 12 |

recorded number of downloads, sample size, median and interquartile range (IQR) for each score (ORCHA, UX, PCA and DP) when separated by assigned download level (1–20). The sample size for download levels varied from 0 to 177 health apps.

Table 4 depicts Spearman correlation that user ratings and downloads had with each other and with each of the quality scores (ORCHA, UX, PCA and DP). For user ratings and downloads, all correlations were weak (<0.2) and not significant with the quality scores. User ratings had weak negative correlations with PCA and DP scores, and weak positive correlation with UX score. UX and DP scores were weakly positively correlated with downloads, while PCA scores were weakly negatively correlated with downloads. User ratings and downloads correlation was .190 and it was statistically significant when adjusted for multiple hypothesis testing with Bonferroni corrected alpha (p < .001).

Table 5 shows the results of MLR, predicting all the assessment scores (separately) with user ratings and download levels. Adjusted R squared was very small for all the scores; however, F-test p-*value*s were statistically significant for UX (p = .005) and DP (p = .003) scores. To make examination of the data more rigorous, the smallest value in the range of values recorded by ORCHA (ORCHA recorded downloads–with plus removed) were also used for comparison.

**Table 3. Downloads details.**

| Recorded downloads | Download levels | Sample size (# apps) | ORCHA score–median (IQR) | UX score–median (IQR) | PCA score–median (IQR) | DP score–median (IQR) |
|---|---|---|---|---|---|---|
| 0+ | 1 | 0 | 0 | 0 | 0 | 0 |
| 1+ | 2 | 0 | 0 | 0 | 0 | 0 |
| 5+ | 3 | 0 | 0 | 0 | 0 | 0 |
| 10+ | 4 | 2 | 47(12) | 64.10(0.90) | 26.85(17.68) | 65.75(6.45) |
| 50+ | 5 | 3 | 54(9) | 70(7.19) | 31.94(10.90) | 69.30(15.08) |
| 100+ | 6 | 30 | 65(19.75) | 73.35(9.61) | 59.30(40.72) | 59.30(16.08) |
| 500+ | 7 | 28 | 66.5(27.25) | 72.20(9.94) | 54.93(46.01) | 63.64(20.06) |
| 1,000+ | 8 | 139 | 61(22) | 74.55(9.95) | 50.89(43.99) | 65(22.53) |
| 5,000+ | 9 | 70 | 62(20.5) | 75.21(8.18) | 53.84(42.59) | 65(21.23) |
| 10,000+ | 10 | 177 | 60(25) | 74.54(10.88) | 47.74(47.05) | 65.39(18.22) |
| 50,000+ | 11 | 70 | 60(22.5) | 77.51(9.94) | 46.72(48.52) | 63.89(15.30) |
| 100,000+ | 12 | 134 | 62(23.75) | 75.21(9.54) | 46.31(45.51) | 66.02(20.98) |
| 500,000+ | 13 | 61 | 66(22) | 74.41(7.73) | 55.67(47.41) | 67.44(21.11) |
| 1,000,000+ | 14 | 72 | 60(19.5) | 75.52(10.11) | 47.95(40.23) | 68.62(13.64) |
| 5,000,000+ | 15 | 31 | 53(20) | 72.95(10.56) | 33.50(44.30) | 66.76(11.34) |
| 10,000,000+ | 16 | 53 | 63(22) | 76.67(8.13) | 50.06(40.49) | 65.74(13.02) |
| 50,000,000+ | 17 | 6 | 61(19.25) | 74.88(8.57) | 41.92(32.56) | 79.05(12.39) |
| 100,000,000+ | 18 | 4 | 53.5(7) | 72.60(2.09) | 24.59(18.71) | 74.64(2.99) |
| 500,000,000+ | 19 | 0 | 0 | 0 | 0 | 0 |
| 1,000,000,000+ | 20 | 1 | 62(0) | 76.67(0) | 41.06(0) | 75.15(0) |

**Table 4. Spearman correlations.** Bonferroni corrected alpha value $.05/9 \approx .006$.

|  | User ratings (p-value) | Downloads* (p-value) |
|---|---|---|
| **ORCHA score** | -.024 (p = .473) | -.011 (p = .747) |
| **UX score** | .010 (p = .759) | .084 (p = .012) |
| **PCA score** | -.043 (p = .207) | -.081 (p = .016) |
| **DP score** | -.020 (p = .545) | .088 (p = .009) |
| **User ratings** | NA | .190 (p < .001) |

*Download levels and ORCHA recorded downloads (plus removed) have the same correlations with other variables.

Figs 1 and 2 depict how scores' medians vary with user ratings and download levels. Independent scores UX, PCA and DP are represented with green, blue and purple lines colours respectively and the dependent ORCHA score is depicted with a red line. The download levels of '1, 2, 3 and 19' are not included since the sample size was 0. Fig 1 in S2 Appendix depicts boxplots for each score per user ratings in the intervals of $>= 1$ and $<2$, $>= 2$ and $<3$, $>= 3$ and $<4$, $>= 4$ and $<= 5$. Figs 2–5 in S2 Appendix depicts each score per download level. Sample size is above each boxplot.

## Discussion

### Principal findings

This study shows that user ratings and number of downloads are inadequate at predicting the quality of health apps. User ratings and download levels demonstrated weak correlations with

**Table 5. MLR results, using both download levels and ORCHA recorded downloads (removed plus).** Bonferroni corrected alpha value $.05/4 \approx .013$.

| Scores | R squared | Adjusted R squared | Model p-value (F-test) | Independent variable |
|---|---|---|---|---|
| ORCHA | .001 | -.002 | .759 | User ratings |
|  |  |  |  | Download level |
| UX | .012 | .010 | .005 | User ratings |
|  |  |  |  | Download level |
| PCA | .006 | .004 | .065 | User ratings |
|  |  |  |  | Download level |
| DP | .013 | .011 | .003 | User ratings |
|  |  |  |  | Download level |
|  |  |  |  |  |
| ORCHA | .001 | -.002 | .782 | User ratings |
|  |  |  |  | ORCHA downloads |
| UX | .001 | -.001 | .704 | User ratings |
|  |  |  |  | ORCHA downloads |
| PCA | .002 | -.0003 | .421 | User ratings |
|  |  |  |  | ORCHA downloads |
| DP | .003 | .001 | .287 | User ratings |
|  |  |  |  | ORCHA downloads |

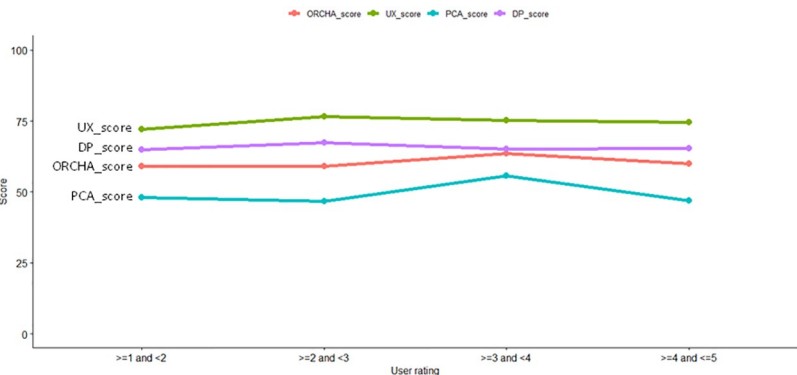

**Fig 1. Median score for each user ratings.**

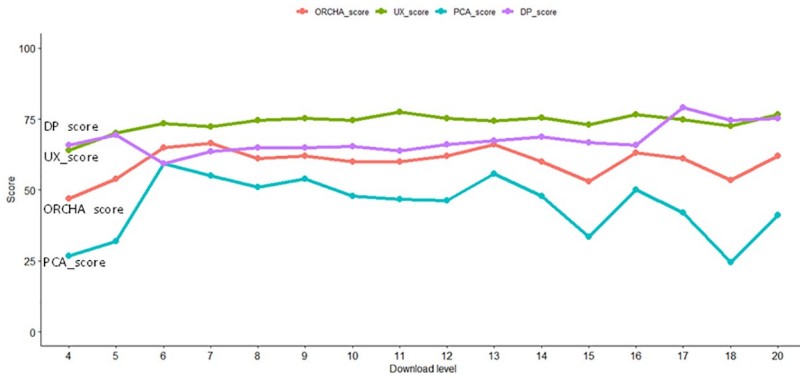

**Fig 2. Median score for each download level.**

all scores (ORCHA, UX, PCA and DP) and each other, as shown in Table 4 (only user ratings and downloads achieved statistically significant correlation with each other when using Bonferroni corrected alpha). Most scores showed a negative correlation with user ratings; UX was the only score that had a positive correlation–albeit weak and not significant. UX and DP scores were positively correlated with download levels, whilst ORCHA and PCA showed a negative correlation with the latter.

The MLR models had low R squared values ($< .02$), as shown in Table 5, meaning that a lot of the variance in the model is unexplained by the model. This further indicates the inadequacy of user ratings and downloads at predicting scores (ORCHA, UX, PCA and DP).

Our findings indicate that user ratings and download levels are not accurate predictors of objective app quality. This suggests that users have difficulty determining, as a basis for their ratings and download decisions, key aspects that contribute to app quality and safety. A potential contributing factor to this may be a lack of digital health literacy. A study from 2021 described digital health literacy and internet connectivity as "super social determinants of health" [25], because they have implications for the wider social determinants of health. A study from 2017, found that individuals who were younger, had more education, reported excellent health, and had a higher income were the main users of health apps [26].

Moreover, our findings are in line with a study from 2022, which provided evidence of a gap between the user ratings and expert ratings from a curated library of over 1,200 apps that

covered physical and mental health [27]. Our results suggest that the cause of this gap may be that health experts look for evidence of clinical quality, utility, privacy, and security that is not considered by users when they rate apps on the iOS and Android app stores. Moreover, users who get their health information, from the internet, rely on search engine results, that may come from unaccredited sources [28]. This indicates that a trusted objective way to judge the quality of health apps is needed.

The study conducted in this paper highlights the need for quality assurance methodologies/tools to accurately determine the quality, safety and compliance of health apps. Our results are in line with the hypothesis that "user ratings and number of downloads are inadequate predictors of user experience, professional/clinical assurance, and data privacy of health apps". The lack of correlation observed between quality assessment tools and user ratings and downloads of health apps suggest that many users use harmful and unsafe health apps, which may partly be due to poor digital health literacy. These issues need to be addressed as departments of health, for example the Food and Drug Administration of the United States [29] or Health and Social Care Northern Ireland [30], are moving towards embracing digital health technologies such as health apps.

## Limitations

This study was limited to Android health apps only, therefore, inclusion of iOS apps, while not expected to be systematically different, may have yielded different findings. User ratings and the number of downloads of health apps included in this study could have changed by the time this study has been published. Additionally, as with any study in digital health, these technologies are highly flexible and subject to change, with updates occurring on a regular basis. Therefore, it is entirely possible that either or both objective compliance of the apps and the number of downloads or user ratings, may have changed since the study began, stressing the need for follow up studies.

OBR is performed by humans and therefore it is entirely possible, although unlikely, that errors can occur in the objective assessment of health apps. The sample size for user ratings ranges (from 8 to 608) and download levels (from 0 to 177) varied widely. Only range of downloads as shown in Table 2 was available for analysis; the exact number of downloads for each health app was unavailable for this study. Which means that precision was not possible, leading to overestimation of download Figs for some and under estimation for others, a natural side effect of transforming continuous data into categorical variables.

## Conclusion

This study shows that online user app ratings and the number of app downloads are inadequate predictors of the quality of the health apps in terms of their user experience, professional/clinical assurance, and data privacy. This indicates the need for quality assurance methodologies/tools to accurately determine the quality, safety and compliance of health apps. It also suggests that the success and uptake of a health app is not based on its quality, which is a worrying prospect given the need for high quality health apps and given the need for digital health literacy amongst citizens. It is important that users self-select high quality health apps as opposed to being misled by user ratings and the popularity of an app.

## Supporting information

**S1 Appendix.**
(DOCX)

**S2 Appendix.**
(DOCX)

**S1 Data.**
(CSV)

## Acknowledgments

We would like to acknowledge the contribution of the many health apps reviewers and developers who worked with ORCHA that allowed for the review of health apps and consented for their data to be used for the purposes of research. Without their contribution and consent this research would not have been possible.

## Author Contributions

**Conceptualization:** Maciej Hyzy, Raymond Bond.

**Data curation:** Maciej Hyzy, Robert Daly.

**Formal analysis:** Maciej Hyzy.

**Investigation:** Maciej Hyzy, Raymond Bond, Maurice Mulvenna, Lu Bai, Simon Leigh.

**Methodology:** Maciej Hyzy, Raymond Bond, Maurice Mulvenna, Simon Leigh.

**Supervision:** Raymond Bond, Maurice Mulvenna, Lu Bai, Simon Leigh.

**Validation:** Raymond Bond, Maurice Mulvenna, Lu Bai, Simon Leigh.

**Visualization:** Maciej Hyzy.

**Writing – original draft:** Maciej Hyzy.

**Writing – review & editing:** Maciej Hyzy, Raymond Bond, Maurice Mulvenna, Lu Bai, Anna-Lena Frey, Jorge Martinez Carracedo, Simon Leigh.

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
