## [Decision Letter · Decision Letter 0]

19 Nov 2023

PONE-D-23-32569Don’t judge a book or health app by its cover: user ratings and downloads are not linked to qualityPLOS ONE

Dear Dr. Hyzy,

Thank you for submitting your manuscript to PLOS ONE. After careful consideration, we feel that it has merit but does not fully meet PLOS ONE’s publication criteria as it currently stands. Therefore, we invite you to submit a revised version of the manuscript that addresses the points raised during the review process.

We look forward to receiving your revised manuscript.

Kind regards,

Kehinde Kazeem Kanmodi, BDS

Academic Editor

PLOS ONE

Journal Requirements:

2. Thank you for providing the following Funding Statement: 

“I have read the journal's policy and the authors of this manuscript have the following competing interests: This study is funded by a DfE Cast award and ORCHA. Simon Leigh, Anna Frey and Robert Daly are/were employees at ORCHA when this study was conducted. Other authors have no conflict of interest to declare.”

We note that one or more of the authors is affiliated with the funding organization, indicating the funder may have had some role in the design, data collection, analysis or preparation of your manuscript for publication; in other words, the funder played an indirect role through the participation of the co-authors.

If the funding organization did not play a role in the study design, data collection and analysis, decision to publish, or preparation of the manuscript and only provided financial support in the form of authors' salaries and/or research materials, please review your statements relating to the author contributions, and ensure you have specifically and accurately indicated the role(s) that these authors had in your study in the Author Contributions section of the online submission form. Please make any necessary amendments directly within this section of the online submission form.  Please also update your Funding Statement to include the following statement: “The funder provided support in the form of salaries for authors [insert relevant initials], but did not have any additional role in the study design, data collection and analysis, decision to publish, or preparation of the manuscript. The specific roles of these authors are articulated in the ‘author contributions’ section.”

If the funding organization did have an additional role, please state and explain that role within your Funding Statement.

Please also provide an updated Competing Interests Statement declaring this commercial affiliation along with any other relevant declarations relating to employment, consultancy, patents, products in development, or marketed products, etc. 

‘This research is done in partnership with ORCHA, a UK-based digital health compliance company. This work is supported by a Northern Ireland DfE CAST award / PhD scholarship. We would like to acknowledge the contribution of the many health apps reviewers and developers who worked with ORCHA that allowed for the review of health apps and consented for their data to be used for the purposes of research”

“This research has been funded by the department for the economy (DfE) in the UK and by company called ORCHA.”

Additional Editor Comments (if provided):

Nil.

Reviewers' comments:

Reviewer's Responses to Questions

**Comments to the Author**

1. Is the manuscript technically sound, and do the data support the conclusions?

Reviewer #1: Yes

Reviewer #2: Yes

2. Has the statistical analysis been performed appropriately and rigorously? 

Reviewer #1: No

Reviewer #2: Yes

3. Have the authors made all data underlying the findings in their manuscript fully available?

Reviewer #1: No

Reviewer #2: Yes

4. Is the manuscript presented in an intelligible fashion and written in standard English?

Reviewer #1: Yes

Reviewer #2: Yes

5. Review Comments to the Author

Reviewer #1: Thank you for sharing this manuscript.

I found your conclusions sound and a valuable addition to the literature.

I do think you need to make some adjustments to your statistical analysis. Primarily, you perform several tests of correlation and several MLRs, and therefore you should be correcting your p-values for multiple hypothesis testing. As this will only serve to make these values less statistically significant, this will not change the conclusions of your study (and indeed will likely make some of the statistically significant correlations no longer significant).

Reviewer #2: Thank you for the opportunity to review your manuscript comparing user ratings to objective assessments of mobile apps for health.

Overall:

Some awkward sentence structures and minor grammatical errors that can easily be corrected.

Introduction

Page 4, lines 88-90: In the sentence: “Therefore, if not sufficiently informed, user’s app choices can result in foregone health benefits if ineffective apps are chosen and/or can pose significant risks to user’s health and privacy.” I wonder if you mean poor health benefits instead of foregone.

Methods

Could you please give a bit more detail about how you determined the download levels from the ranges?

It may also be useful to your readers to have a brief explanation about why you chose the analyses described.

Results

Page 11, Lines 224-225: You may want to consider using a different method for differentiating between your scores on Figures 1 and 2 as these colours may be difficult for someone who is colour-blind to read.

6. PLOS authors have the option to publish the peer review history of their article (what does this mean?). If published, this will include your full peer review and any attached files.

Reviewer #1: No

Reviewer #2: No

---

## [Author Response · Author response to Decision Letter 0]

19 Dec 2023

Dear Editor and Reviewers,

Thank you for the feedback and the opportunity to improve our manuscript. We have now addressed reviewers’ comments. Below is our point-by-point response to the comments.

Reviewer #1: Thank you for sharing this manuscript.

I found your conclusions sound and a valuable addition to the literature.

I do think you need to make some adjustments to your statistical analysis. Primarily, you perform several tests of correlation and several MLRs, and therefore you should be correcting your p-values for multiple hypothesis testing. As this will only serve to make these values less statistically significant, this will not change the conclusions of your study (and indeed will likely make some of the statistically significant correlations no longer significant).

Reviewer 1 response A1: We acknowledge that the p-values should be corrected for multiple hypothesis testing. We have now revised the analysis and included Bonferroni corrected alpha values.

Reviewer #2: Thank you for the opportunity to review your manuscript comparing user ratings to objective assessments of mobile apps for health. Overall: Some awkward sentence structures and minor grammatical errors that can easily be corrected.

Introduction: Page 4, lines 88-90: In the sentence: “Therefore, if not sufficiently informed, user’s app choices can result in foregone health benefits if ineffective apps are chosen and/or can pose significant risks to user’s health and privacy.” I wonder if you mean poor health benefits instead of foregone.

Reviewer 2 response A1: We have now reworded the sentence to “Therefore, if not sufficiently informed, user’s app choices can result in poor health benefits if ineffective apps are chosen and/or pose significant risks to user’s health and privacy.”

Methods: Could you please give a bit more detail about how you determined the download levels from the ranges? It may also be useful to your readers to have a brief explanation about why you chose the analyses described.

Reviewer 2 response A2: The following paragraphs were modified in the manuscript. 

“Data was accessed and analysed between ‎July and December 2022. We carried out secondary data analyses of this ORCHA dataset, using R studio and R programming language. Spearman correlations were used to examine how correlated ORHCA, UX, PCA and DP scores are with user ratings (a 1-5 rating) and number of downloads. The number of downloads variable was converted into download levels, as only download ranges, not exact numbers of downloads, were available. There were 20 ranges of downloads, and each was assigned a download level going from 1 (the smallest) to 20 (the highest). For the analysis the smallest value in each of the 20 ranges was also used as an alternative to the download levels. This was done to improve rigour of the analysis by using two approaches to estimate number of downloads from the available range of downloads.”

“Multiple linear regression (MLR) was used to model the relationship between app quality scores and the apps’ user ratings and downloads. R squared and adjusted R squared metrics were used to measure the fitness of the models. For all statistical tests, a p-value <.013 (Bonferroni-corrected for multiple hypothesis testing) was considered statistically significant. If there are any links among user ratings and downloads, and quality scores they should be revealed by spearman correlations and/or MLR.”

Results: Page 11, Lines 224-225: You may want to consider using a different method for differentiating between your scores on Figures 1 and 2 as these colours may be difficult for someone who is colour-blind to read.

Reviewer 2 response A3: We acknowledge that people with colour-blindness may not be able to differentiate the scores in the figures. Hence, in addition to using colours, we have now labelled figures 1 and 2 with text.

---

## [Decision Letter · Decision Letter 1]

2 Feb 2024

Don’t judge a book or health app by its cover: user ratings and downloads are not linked to quality

PONE-D-23-32569R1

Dear Dr. Hyzy,

We’re pleased to inform you that your manuscript has been judged scientifically suitable for publication and will be formally accepted for publication once it meets all outstanding technical requirements.

Kind regards,

Kehinde Kazeem Kanmodi, BDS

Academic Editor

PLOS ONE

Additional Editor Comments (optional):

Nil.

Reviewers' comments:

Reviewer's Responses to Questions

**Comments to the Author**

1. If the authors have adequately addressed your comments raised in a previous round of review and you feel that this manuscript is now acceptable for publication, you may indicate that here to bypass the “Comments to the Author” section, enter your conflict of interest statement in the “Confidential to Editor” section, and submit your "Accept" recommendation.

Reviewer #2: All comments have been addressed

2. Is the manuscript technically sound, and do the data support the conclusions?

Reviewer #2: (No Response)

3. Has the statistical analysis been performed appropriately and rigorously? 

Reviewer #2: (No Response)

4. Have the authors made all data underlying the findings in their manuscript fully available?

Reviewer #2: (No Response)

5. Is the manuscript presented in an intelligible fashion and written in standard English?

Reviewer #2: (No Response)

6. Review Comments to the Author

Reviewer #2: (No Response)

7. PLOS authors have the option to publish the peer review history of their article (what does this mean?). If published, this will include your full peer review and any attached files.

Reviewer #2: No

---

## [Editor Report · Acceptance letter]

22 Feb 2024

PONE-D-23-32569R1 

PLOS ONE

Dear Dr. Hyzy, 

I'm pleased to inform you that your manuscript has been deemed suitable for publication in PLOS ONE. Congratulations! Your manuscript is now being handed over to our production team.

Kind regards, 

on behalf of

Dr. Kehinde Kazeem Kanmodi 

Academic Editor

PLOS ONE